# Plant-Derived Natural Products: A Source for Drug Discovery and Development

**Noureddine Chaachouay** [1,*] and Lahcen Zidane [2]

1 Agri-Food and Health Laboratory (AFHL), École Supérieure Normale, Hassan First University, P.O. Box 382, Settat 26000, Morocco
2 Plant, Animal Productions and Agro-Industry Laboratory, Department of Biology, Faculty of Sciences, Ibn Tofail University, P.O. Box 133, Kenitra 14000, Morocco
* Correspondence: noureddine.chaachouay@uhp.ac.ma; Tel.: +212-677-488-621

**Abstract:** For thousands of years, nature has been a source of medical substances, and an astounding numeral of contemporary remedies have been identified from natural origins. Plants have long been used as folk herbal medicines to treat various disorders, and their different natural products have inspired the design, discovery, and development of new drugs. With the invention of recent molecular targets based on proteins, there is a growing need for fresh chemical diversification in screening. Natural products will play a vital part in supplying this need via the continuous exploration of global biodiversity, the majority of which remains unexplored. Even though drug discovery from medicinal plants remains an important source of novel therapeutic leads, various hurdles exist, including identifying and executing suitable high-throughput screening bioassays, scaling up the supply of bioactive molecules, and acquiring plant materials. Investigating these natural resources takes multi-disciplinary, nationwide, and global partnerships in design, synthesis, discovery, and drug development techniques. This review article discusses current advancements and future approaches for discovering natural items such as health- and wellness-promoting remedies. It also summarizes strategies to unify the therapeutic use of plant-derived natural products worldwide to support future drug discoveries derived from plant sources.

**Keywords:** drug discovery; medicinal plant; medicine; natural products; pharmacy

## 1. Introduction

Medicinal and aromatic plants, especially those with ethnopharmacological uses, have been utilized as a natural source of remedies and healthcare for millennia [1–3]. Initially, these popular medications were primitive formulations such as powders, tinctures, macerations, teas, infusions, percolation products, poultices, decoctions, tinctures, inhalations, and other herbal preparations [4–6]. The precise dose of the plant and the mode of administration for specific diseases have been transmitted by oral tradition from one generation to another, and traditional pharmacopeias eventually documented knowledge of medicinal plants [4,7,8]. Multiple disciplines of study and diverse investigation methods have been included in drug discovery from medicinal plants. Botanists, ethnobotanists, ethnopharmacologists, and plant ecologists often gather and identify the plants of interest [9]. New technological developments enable plants to be transformed into "factories" that create natural medical compounds for use in the production of biotech pharmaceuticals, medications, and treatments [10]. The application of plants as drugs has recently required the separation of active ingredients, starting with the early-19th-century isolation of morphine from *Papaver somniferum* [11,12]. The identification of early pharmaceuticals, such as digitoxin, cocaine, pilocarpine, codeine, and quinine, that are derived from medicinal plants marked a remarkable achievement in the field of medicine [13]. These compounds have been separated and analyzed for their medicinal characteristics and are still acknowledged for their therapeutic uses in the present day. Furthermore, apart from these initial findings,

several additional molecules originating from plants have been identified in more recent times. These compounds have undergone extensive research and development and have subsequently been commercialized as pharmaceutical medications [14]. Scientists' investigation of medicinal plants has been crucial in uncovering early drugs, each possessing distinct pharmacological characteristics. Paclitaxel, obtained from *Taxus brevifolia*, is used in the treatment of lung, ovarian, and breast cancer. The compound artemisinin, derived from the traditional Chinese herb *Artemisia annua*, is used to battle malaria, which is resistant to many drugs [15]. Silymarin, derived from the seeds of *Silybum marianum*, is used for the treatment of hepatic disorders. Digitoxin, extracted from the foxglove plant, has been used in the management of cardiac ailments, specifically, congestive heart failure [16]. Cocaine, first known for its anesthetic effects, was traditionally used for local anesthesia and constriction of blood vessels. Pilocarpine, derived from the jaborandi plant, has been used in medicine for its ability to stimulate salivation and perspiration [17]. Codeine, an opioid alkaloid derived from the opium poppy, gained significant popularity due to its pain-relieving and cough-suppressing qualities. Quinine, derived from the bark of the cinchona tree, played a crucial role in the fight against malaria [9,18,19]. Notably, several of these early drugs, including digitoxin and quinine, are still employed in contemporary medicine, illustrating the continuing significance of these plant-derived substances [9]. In addition, scientists' continuous endeavors to separate and describe pharmacologically active substances from medicinal plants have resulted in the identification of supplementary compounds with therapeutic promise. The procedure includes a thorough study to comprehend the chemical structures, modes of action, and possible medicinal uses of these substances. This continuous investigation highlights the significance of nature as a significant source of bioactive compounds that continue to aid in the creation of new medications and treatments in modern medicine. To date, drug development approaches have been used to standardize herbal remedies to identify analytical marker biomolecules [20]. Plant-made pharmaceuticals result from the creative application of biotechnology to plants to make drugs derived from natural products that the medical profession can employ to fight life-threatening ailments, such as asthma, influenza, cancer, tuberculosis, diabetes mellitus, coronary artery disease, and diarrhea [21].

Developing drugs using plant-based pharmaceutical techniques offers an efficient, cost-effective, and safe alternative to conventional procedures using animal cell cultures or microbial fermentation. Therefore, drugs derived from natural compounds in plants can offer patients greater and quicker access to medications [22,23]. The most remarkable characteristic of natural products concerning their enduring significance in drug development is their mostly unexplored structural diversity. This research summarizes an overview of plant-derived natural products as drug discovery and development candidates.

## 2. Methodology

The data on plant-derived natural products used as a source for drug discovery and development were obtained through literature publications using different scientific literature search engines, including Springer, Wiley Online, PubMed, Google Scholar, ResearchGate, ScienceDirect, Taylor & Francis, Web of Science, MDPI, Academia.edu, Bentham, Thieme, Scopus, SpringerLink, and SciFinder. Other literature references were also used, such as academic library books and newspapers. The terms "Plant-derived natural products", "Drug discovery", and " Medicinal plant" were used in the investigation. The resources and books were chosen according to the subject covered. Approximately 245 papers in the literature were reviewed; only 224 references were included in this analysis. We did not include reports or literature on synthetic chemicals used as a source for drug discovery and development. Literature on plant-derived natural products used as suitable precursors for drug discovery and development was included. These papers were carefully evaluated, critically analyzed and structured with accurate information.

### 3. Historical Significance

The historical significance of herbal medicine exemplifies the enduring relationship between humans and the natural world in the pursuit of health and well-being [24]. Throughout history, several cultures from across the globe have acknowledged and used the therapeutic qualities of plants. Herbal treatments have been used since ancient times, as shown by archeological findings that suggest the usage of medicinal herbs as early as the Paleolithic era, some 60,000 years ago [25,26]. The Sumerians, who kept lists of plants, left written records of herbal treatments that may be traced back over 5000 years [27]. Subsequently, the use of herbs has seen fluctuations in popularity within the medical domain; there were several instances when ancient societies such as the Egyptians, Greeks, and Romans heavily relied on herbal medicines for medicinal purposes [28]. These traditions have not only endured throughout time but also have significantly shaped the advancement of contemporary pharmacology and healthcare practices [29]. Exploring significant milestones in the historical use of plant-based treatments offers a comprehensive viewpoint on this long-lasting connection:

#### 3.1. Ancient Egyptian Healing Practices (Circa 1500 BCE)

The ancient Egyptians are revered for their pioneering use of medicinal plants. Their civilization's written records, dating back to 1500 BCE, include extensive documentation of herbal remedies [25]. These records provide a thorough understanding of the diverse range of botanicals they use for medicinal purposes. Their expertise and methodologies serve as the basis for several contemporary herbal treatments and medical procedures [30]. The Ebers Papyrus is an ancient Egyptian medical manuscript that has a vast amount of information on herbal medicines and other medicinal therapies. It is estimated to have originated about 1550 BCE and is among the most ancient medical texts on record. The papyrus has more than 700 enchanting incantations and traditional treatments, a significant portion of which are taken from botanical sources and herbal extracts. The manuscript is widely regarded as one of the most comprehensive and well-preserved documents of ancient Egyptian medicine in existence today [31].

#### 3.2. Chinese Herbal Medicine (Ancient Times)

China boasts an extensive heritage of herbal medicine, which has been diligently employed for millennia. The foundation of this custom rests in the tenets of traditional Chinese medicine (TCM), an empathetic system that uses botanical remedies to reinstate equilibrium and foster holistic well-being [32,33]. Chinese herbal medicine is founded upon a comprehensive understanding of numerous plant, mineral, and animal substances, a considerable number of which continue to hold substantial relevance in modern TCM [18]. TCM practitioners support good health and enhance organ function through the use of botanicals and herbal formulas. A comprehension of the fundamental nature of different herbal constituents empowers the TCM practitioner to generate a therapeutic impact that transcends the herbs' chemical makeup and physical attributes. Herbal formulations from China, some of which have been utilized for over 2200 years, consist of components selected for their complementary functions. In contrast to the individualistic approach prevalent in Western medicine, TCM frequently involves the combination of botanicals for the purpose of eliciting a synergistic effect [34,35].

#### 3.3. Plant-Derived Products in the Traditional Pharmacopeias

Secondary metabolites are organic substances synthesized by plants that do not directly participate in the fundamental processes of growth, development, or reproduction [36]. However, these metabolites often fulfill ecological roles, such as acting as a defense mechanism against herbivores, attracting pollinators, or engaging in competition with other plants [37,38]. Several secondary metabolites have been used for their therapeutic qualities and are essential constituents of traditional pharmacopeias. As an example, alkaloids, which belong to a group of nitrogen-containing compounds, are found in numerous plant

species and demonstrate an extensive array of pharmacological effects. Morphine derived from the opium poppy, quinine produced by the cinchona tree, and caffeine originating from coffee beans are some examples [39]. Traditional medicine has employed these compounds for their respective analgesic, antimalarial, and stimulant properties for centuries [40]. These chemicals have been used for many centuries in traditional medicine due to their pain-relieving, antimalarial, and stimulating effects, respectively [41,42].

Traditional pharmacopeias, which are based on the wisdom and methods of native societies, often depend on plant-based substances that are rich in secondary metabolites [43]. Herbal treatments and formulations derived from traditional knowledge have been transmitted over centuries and still have a prominent position in healthcare practices across many cultures [44,45]. These formulas are regarded as having not only medicinal but also cultural significance. Traditional Chinese medicine encompasses a wide range of plant-based medicines, which include herbs that contain secondary metabolites, including flavonoids, terpenoids, and saponins [33]. Similarly, Ayurveda, the indigenous medical system of India, primarily utilizes plant-derived formulations, including secondary metabolites such as alkaloids and polyphenols [46]. The incorporation of secondary metabolites originating from plants into traditional pharmacopeias underscores the significance of biodiversity in promoting human health and the utility of natural substances for the discovery and advancement of drugs.

### 3.4. Quinine: A Malaria Breakthrough from Cinchona (17th Century)

For centuries, malaria, an extremely lethal illness induced by the Plasmodium parasite, has afflicted the human race. As a fever remedy, the bark of the cinchona tree (*Cinchona officinalis*) was utilized by the indigenous people of South America [47]. Quinine, the bioactive compound found in cinchona bark, was recognized as an exceptionally productive malaria remedy during the 17th century [48]. This finding represented a significant advancement in managing the ailment; quinine persisted as a pivotal anti-malarial medication for centuries until more contemporary therapies were devised [49].

The historical importance of plant-based medicine demonstrates the lasting wisdom of ancient civilizations in acknowledging the therapeutic capabilities of the natural environment [50]. These first techniques established the groundwork for the evolution of contemporary pharmacology and have played a role in the progress of healthcare [51]. Furthermore, they emphasize the need to conserve and honor our natural resources while simultaneously investigating the medicinal possibilities of plant-based natural products in the modern realm of pharmaceutical research and advancement.

### 3.5. Aspirin: A Gift from the Willow Tree (19th Century)

The historical use of botanical remedies has had a substantial impact on the advancement of contemporary pharmacology. Aspirin, an extensively used analgesic on a worldwide scale, serves as a prominent illustration of this phenomenon. Aspirin was first extracted from the bark of the *Salix alba* species, often known as the willow tree [52]. This natural treatment underwent a process of synthesis, resulting in its transformation into a synthetic form. As a result, it has become well recognized and is used worldwide as a fundamental component in pain management. Hence, the natural world's collection of medical substances has significantly influenced the current pharmaceutical industry, with aspirin serving as a prime example of the shift from traditional herbal treatments to artificially created medicinal chemicals [52]. The Greeks and Egyptians, among other ancient civilizations, used willow bark as an analgesic. During the 19th century, scientists effectively extracted the active ingredient, acetylsalicylic acid, which brought about a significant transformation in the treatment of pain and control of inflammation [53].

## 4. Importance of Plant-Derived Natural Products in Drug Discovery

Historically, natural compounds and their structural analogs have contributed significantly to pharmacotherapy, particularly for infectious disorders [54]. In addition to the

physiologically active bioactive molecules generated from plants that have been discovered to have direct therapeutic use as drug substances, several additional natural bioactive chemicals have proved effective as "leads" or model molecules for drug synthesis or semisynthesis [55,56].

Recently, several scientific technical and advancements, including engineering methodologies, genome mining, enhanced analytical tools, and microbial culture advancements, have helped to overcome these issues and provide new possibilities [57]. These advancements are evident in the complexness of bioassays used in medicinal-plant drug development. They also give "mode of action" data at the molecular scale in a timely and precise manner [56,58]. Accordingly, there is a resurgence of interest in natural compounds as potential therapeutic leads, notably for treating antimicrobial resistance [59].

Today, researchers specializing in ethnobotany, pharmacognosy, pharmaceutics, medicinal chemistry, botany, taxonomy, organic chemistry, molecular biology, biochemistry, microbiology, pharmacology, and plant ecology may collaborate to discover new chemical components in medicinal plants [60,61]. Standard procedures in plant natural product separation chemistry for drug discovery may be divided into five stages: species collection, extraction, chemical separation, structural identification, and bioassays [62,63]. These procedures are applied with combinatorial and synthetic chemistry initiatives, computational modeling and chemical informatics research, and unique high-throughput screening approaches in pharmaceutical and other laboratories with significant assets [64–66]. Plant-derived natural products remain crucial in drug discovery and development, providing a plentiful supply of bioactive molecules with various characteristics. Several convincing factors highlight the lasting importance of natural compounds produced by plants in this context.

### 4.1. Chemical Diversity

The distinctive characteristic of plant-derived natural products is their extensive range of secondary metabolites, which contributes to their outstanding chemical diversity. The metabolites include a variety of compounds, such as alkaloids, flavonoids, terpenoids, and phenolic substances, among others. The wide range of chemical compounds found in natural products generated from plants has served as a valuable source of inspiration for the process of identifying and creating new drugs. Several of these substances have been used in traditional medicine for several years and are now being investigated for their medicinal capacity. The discovery and analysis of these substances have resulted in the development of novel medications and therapies for a diverse array of illnesses and ailments [55,67]. These compounds exhibit distinct chemical structures and a wide range of pharmacological effects. The content of plant-derived natural compounds makes them a perfect source for discovering prospective medication candidates. Pharmaceutical researchers have the opportunity to investigate this extensive collection of substances to uncover new molecules that have the potential to be used in therapy.

### 4.2. Evolutionary Adaptations

Plants have developed complex chemical defense systems over millions of years to safeguard themselves against many dangers, including herbivores, viruses, and environmental stresses. Several of these protective chemicals have exceptional biological activity and a significant level of selectivity in targeting specific biological processes [68]. For example, several plant chemicals can interfere with the eating or reproductive habits of herbivores. Within the realm of plant-produced molecules, certain substances exhibit bioactivity, indicating their ability to engage with biological systems, including those of other species. Scientists diligently investigate these bioactive chemicals as potential options for innovative treatments. The drug development process includes a rigorous examination of the mechanisms of action and therapeutic uses of these molecules [69]. This generally involves the operation of isolating, purifying, and characterizing these compounds in order to uncover their chemical structures and comprehend their biological impacts. Through

an in-depth exploration of the complexities of plant chemistry, scientists strive to use the capabilities of these substances in order to develop precise medications that target certain illnesses or situations [70]. Instances such as aspirin and Taxol show the triumph of converting plant defensive adaptations into potent drugs, emphasizing the importance of this method in pharmaceutical advancement [53,71]. Furthermore, the sustainable and organic characteristics of plant-based chemicals are in line with the current focus on ecologically responsible pharmaceutical research, using the renewable resources provided by plants [72].

*4.3. Traditional Knowledge*

Indigenous and traditional systems of medicine have amassed a vast amount of information about the medicinal qualities of plants over several generations. These systems, based on local knowledge and practical experience, have long acknowledged the therapeutic properties of different plant-derived treatments [73]. Traditional knowledge often acts as a priceless foundation for contemporary endeavors in drug development. By collaborating with indigenous populations and investigating the customary use of plants, researchers might discern auspicious prospects for therapeutic development [74]. The integration of conventional knowledge with modern scientific techniques helps optimize the process of discovering novel medications.

Additional plant-derived natural products of medical significance that were earlier acquired from herbal origins but are currently manufactured economically mainly via synthesis include atropine, cocaine, ephedrine, colchicine, caffeine, digitoxin, morphine, quinine, scopolamine, theobromine, and Taxol. For example, atropine derivatives are a substantial class of medications derived from a natural-product lead ingredient. Similarly, synthetic analogs of atropine are responsible for more authorized medications than any other plant bioactive compound [75]. In addition, artemisinin is a natural substance with many semisynthetic derivatives. The fundamental motivation for developing these compounds was to address artemisinin's limited water solubility and short plasma half-life. The World Health Organization officially prescribes artemisinin-based oral combination therapies for malaria [76,77]. Additionally, salicylic acid, a salicin-like molecule, became widely accessible once its simple production was developed. Two further semisynthetic derivatives of salicylic acid that are therapeutically noteworthy today are mesalamine and *p*-aminosalicylic acid, which are recommended for the treatment of ulcerative colitis and TB, respectively [66,78–80]. These examples demonstrate the continued utility and significance of natural products generated from plants as template chemicals for current drug discovery and development.

In general, the relevance of plant-derived natural products in medication discovery and development is undeniable owing to their chemical variety, their evolutionary adaptations, and the abundance of traditional knowledge surrounding their usage. These natural molecules have the potential to deliver unique answers to a wide variety of medical problems, making them an enduring and vital resource in the search for novel medications and cures. Their contribution to the pharmaceutical industry shows how the natural world and contemporary science may work together to improve human health and well-being.

**5. Methods for Discovery**

Discovering and creating therapeutics from natural substances found in plants is an intricate and diverse journey that encompasses several essential stages, each of which has a vital part in transforming the potential of botanical chemicals into successful medications. Below is a thorough analysis of these fundamental techniques for exploration (Figure 1).

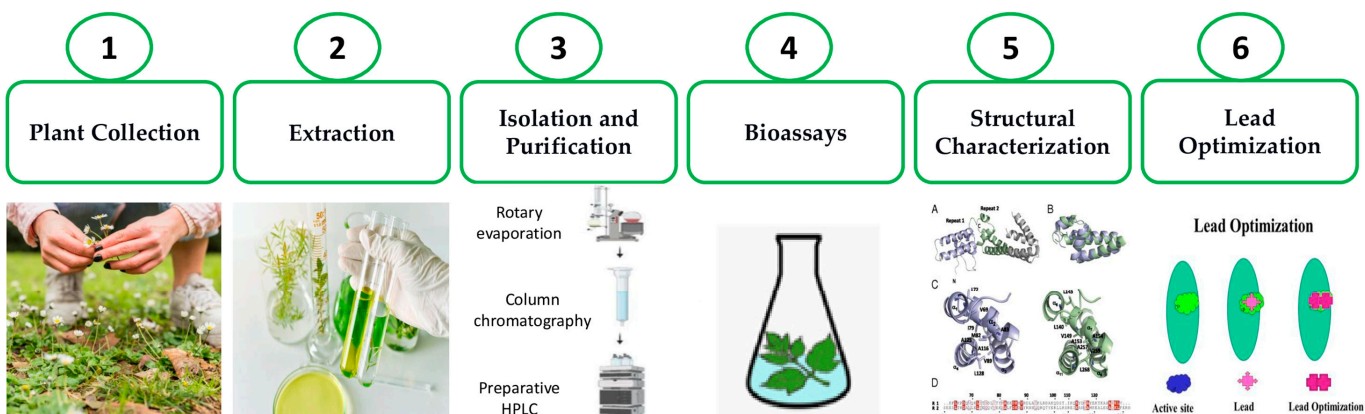

**Figure 1.** Various stages of the drug discovery process from natural products (1-Plant collection, 2-Extraction, 3-Isolation and purification, 4-Bioassays, 5-Structural characterization, 6-Lead optimization).

### 5.1. Plant Collection

During the first stages of drug development, the involvement of botanical specialists and ethnobotanists is crucial in the identification and collection of plant species that may possess therapeutic characteristics [81,82]. Plant selection may be influenced by several sources of information, such as traditional knowledge, which often originates from indigenous and local people. Additionally, ethnobotanical research and field surveys may be carried out to identify potential plant candidates with promising qualities. The collection phase is an essential first step in obtaining the wide range of bioactive chemicals in the plant kingdom [81].

### 5.2. Extraction

After the plant material has been gathered, the subsequent stage extracts bioactive chemicals from the plant matrix. Various methodologies may be used to achieve this objective, contingent upon the chemical composition of the chemicals and the characteristics of the plant material [83]. Typical techniques for extracting substances include maceration, where the plant material is soaked in a suitable solvent; Soxhlet extraction, which provides continuous extraction using a solvent; and supercritical fluid extraction, which utilizes supercritical fluids to extract chemicals selectively [84].

Eco-extraction methods, commonly referred to as green extraction techniques, prioritize the reduction of environmental harm and the promotion of sustainability in the process of extracting bioactive components from plant matrices [85]. Supercritical fluid extraction is a commonly used method that uses $CO_2$ as a solvent at supercritical conditions. This allows it to exhibit the characteristics of both a liquid and a gas [86]. Supercritical $CO_2$ is characterized by its lack of toxicity and flammability and by its ease of removal, making it an environmentally beneficial substitute for conventional organic solvents [87]. Pressurized hot water extraction is an additional technique of eco-extraction that uses water as a solvent under high pressure and temperature [88]. This method minimizes the need for organic solvents and has the additional benefit of maintaining the stability of heat-sensitive bioactive substances.

Investigators are investigating alternate options to traditional organic solvents in order to find sustainable and bio-based solvents that do not have detrimental effects on the environment and human health [89]. An encouraging alternative is the use of ionic liquids, which are compounds that exist as salts in a liquid form at relatively low temperatures. Ionic liquids provide characteristics such as low volatility, excellent heat stability, and adjustable qualities, which make them appropriate for a wide range of extraction procedures [90]. Deep eutectic solvents are a novel category of bio-based solvents that consist of natural and renewable chemicals, including organic acids and amines [91]. Dioctyl sulfosuccinates

are regarded as environmentally friendly options because of their capacity to decompose naturally, their low degree of harmfulness, and their capability to dissolve a diverse array of biologically active substances [92]. The use of eco-extraction techniques and bio-based solvents facilitates the advancement of sustainable practices in extracting bioactive compounds from plant matrices. These developments contribute to a more sustainable and ethical method of extracting valuable chemicals from plants by minimizing the negative impact on the environment and lowering health hazards.

### 5.3. Isolation and Purification

The plant material that is obtained usually consists of a combination of different chemicals, including bioactive and non-bioactive elements [93]. To separate the specific active components, the mixture undergoes fractionation and purification procedures [94]. This may include using methodologies such as column chromatography, which segregates compounds according to their chemical characteristics, and HPLC (high-performance liquid chromatography), a high-resolution approach for segregating and measuring molecules. Additional techniques, such as solvent partitioning and crystallization, may also purify the desired molecules [95].

### 5.4. Bioassays

Following the identification and purification of compounds, the subsequent pivotal stage involves assessing their pharmacological efficacy. Both in vitro and in vivo bioassays are used to evaluate the effectiveness, safety, and probable mechanisms of action of the drugs [96]. In vitro experiments are performed under controlled laboratory conditions, often using cell cultures or isolated biochemical systems, to assess the impact of substances at a cellular or molecular level [97]. In contrast, in vivo experiments include evaluating the chemicals in live beings to provide insights into their physiological and toxicological characteristics.

### 5.5. Structural Characterization

Establishing the chemical composition of intriguing substances is crucial for comprehending their characteristics and potential in pharmaceutical advancement [98]. Commonly used methods for determining the chemical structure of isolated substances include nuclear magnetic resonance (NMR) spectroscopy and mass spectrometry [99]. These approaches provide comprehensive insights into the spatial organization of atoms and functional groups inside the molecules.

### 5.6. Lead Optimization

After identifying a lead chemical with potential pharmacological action, the drug development process progresses to lead optimization. Medicinal chemists and pharmacologists collaborate to enhance the characteristics of the primary chemical [100]. This may include altering the chemical composition to improve potency, selectivity, and pharmacokinetic characteristics. The objective is to create a molecule that not only has therapeutic promise but also fulfills the rigorous requirements for a safe and productive medication [101].

The process of drug development using plant-derived natural compounds, starting with plant collection and leading to lead optimization, is a systematic and well-organized trip [102,103]. This systematic approach integrates scientific rigor with conventional knowledge, facilitating the exploration of novel and inventive pharmaceuticals derived from a diverse range of natural resources. Every stage in this process is crucial for effectively transforming herbal cures into life-saving drugs.

## 6. Biodiversity of Medicinal Plants

The term "biodiversity" refers to the universality of all life forms, including microbes, animals, and plants, as well as the range of environments in which they reside. It

also includes variances in genetic makeup and the different ecosystems that result from them [104,105]. There are three distinct categories of biodiversity: species diversity, ecological diversity, and genetic diversity. Biodiversity is connected to chemodiversity, which plays a vital role in discovering and developing pharmaceuticals and chemotherapeutic drugs [106,107].

Plant biodiversity consists of all plant species on Earth. Within a specific natural habitat or "biotope", which translates to "place of life", plant and animal species live and interact to build a healthy ecosystem [108]. In their natural condition, plants cover almost the whole planet's surface. Only the highest summits and pack ice are free of vegetation. However, depending on temperature and latitude, the vegetation changes significantly [109]. There are roughly 391,000 taxa of vascular plant species that have been identified, with around 369,000 species being flowering plants, according to a study from the Royal Botanic Gardens, Kew, in the United Kingdom [110]. Additionally, around 2000 new plant species are discovered or described each. Numerous newly found species are already threatened with extinction.

In 2015, for example, botanists reported *Gilbertiodendron maximum* Burgt & Wieringa, a severely endangered, 105-metric-ton, enormous tree located in the Cameroon–Congo jungle [110]. Additionally, researchers discovered *Oberholzeria etendekaensis*, a succulent shrublet that is both a new species and a new genus. It is a rare species known only from a single place in Namibia, where 30 specimens have been seen [111]. Brazil, China, and Australia are the top three suppliers of newly discovered taxa each year. Furthermore, Brazil has more seed plants than any other country in the world, and our grasp of its flora is developing at a "record-breaking" pace [110].

Although specific areas across the world still have a wide variety of plant species, including some that are uncommon and endangered, it is essential to acknowledge the lack of legal protection for these plants. Recent data indicate a troubling imbalance, showing that even though these places are ecologically important and potentially delicate, only a small percentage of plant species are protected by legal measures [112]. The absence of regulations in this area exposes a wide range of valuable plants to the risks of exploitation, damage to their natural habitats, and uncontrolled human activities [113]. The lack of adequate legal safeguards not only endangers the survival of these uncommon plants but also emphasizes the pressing requirement for improved conservation efforts, policy changes, and global cooperation to tackle the imminent dangers and guarantee the sustainable preservation of these crucial ecosystems. Overall, investigators have identified 1771 critical plant sites worldwide that need immediate conservation intervention [110]. The primary risks include wildfires, habitat degradation, agriculture, construction, and deforestation, while climate change has a minor effect [114,115]. As a result, sustainable conservation programs are required to conserve these environments and their biodiversity.

## 7. Diversity of Plant-Derived Compounds

Plants can produce a diverse array of chemical compounds, each possessing distinct qualities and serving specific functions. These substances are often known as secondary metabolites and do not directly participate in the regular growth, development, or reproduction of the plant [116]. Instead, plants manufacture these compounds for several purposes, including protection against predators and diseases, attraction of pollinators, and communication with other species [117,118]. These molecules include a wide range of groups, such as alkaloids, flavonoids, terpenoids, and polyphenols, and they often have notable therapeutic characteristics.

Alkaloids are nitrogen-containing chemicals in different plant species and may have significant physiological effects. Morphine, obtained from the opium poppy, is a prime example of an alkaloid used for its pain-relieving characteristics [119,120]. Conversely, flavonoids are recognized for their antioxidant and anti-inflammatory properties. They are often present in fruits, vegetables, and other plant sources, contributing to their qualities that promote good health [121]. Terpenoids, such as menthol and Taxol, display

many biological actions and are often used in traditional medicine and pharmaceutical research [122,123]. Polyphenols, found in foods such as green tea, red wine, and dark chocolate, have attracted interest due to their potential health advantages, such as their antioxidant and antiaging characteristics [124,125].

Many fruits and vegetables contain a wide range of flavonoids, which are chemicals known for their colorful hues and many biological functions [126]. Citrus fruits, such as oranges and grapefruits, are rich in flavonoids such as hesperidin and naringin, which are well known for their antioxidant characteristics and possible advantages for cardiovascular health [127]. Berries, such as blueberries and strawberries, include anthocyanins, which are flavonoids known for their anti-inflammatory properties and beneficial effects on cognitive function [128]. Apples include quercetin, a flavonoid compound that has antioxidant and anti-inflammatory properties, which promote cardiovascular well-being [129]. Grapes, mainly when consumed as red wine, contain resveratrol, a flavonoid that is linked to safeguarding the cardiovascular system and has putative anti-aging properties [130]. Green tea contains a significant amount of epigallocatechin gallate (EGCG), which is a flavonoid known for its antioxidant qualities and has been extensively researched for its ability to prevent cancer [131]. Combined, the flavonoids demonstrate several biological functions, including antioxidant, anti-inflammatory, anticancer, and cardiovascular-protective actions. This emphasizes the potential of a diet abundant in a variety of fruits and vegetables to promote good health.

The extensive and irreplaceable resource for the investigation and advancement of prospective therapeutic candidates lies in the abundant chemical variety of these and other chemicals originating from plants. Scientists and researchers are actively exploring the capabilities of these natural substances to tackle various medical diseases and develop groundbreaking pharmaceuticals to improve world health.

## 8. The Place of Plants in the Traditional Pharmacopeia

The use of plants as remedies predates recorded human history, and some of the oldest written documents from Mesopotamia, Egypt, India, China, Mesoamerica, and Sumerian discuss the issue [132–134]. The medicinal plant is an essential element of popular medical approaches worldwide in treating and preventing human/livestock disorders [135–137]. Aside from that, these plants play an essential function in the evolution of human civilizations worldwide. Nearly every civilization in the world has a body of knowledge about the medicinal benefits of native plants. Moreover, interactions between various cultures have increased the pharmacopeia of one group due to the acceptance of plants utilized by the other to treat several ailments [138,139]. Later, ancient practitioners investigated medicinal plants' qualities and therapeutic benefits in depth and experimentally documented them, creating the basis of ancient medical knowledge [140].

At present, herbal medicine is used in every region of the world. Its economic significance is expanding quickly, mainly because medicinal plants have achieved respectable standing in recent years, particularly in underdeveloped nations where modern health care is restricted and medicinal plants constitute the only available therapy. These emerging countries share key characteristics, such as severely low education levels, poor communications, limited resources, enormous distances between population centers, and communal and individual poverty [141,142]I. Traditional herbal medicines suggest a long history of use, valid for various items marketed as traditional medicines [143]. Furthermore, herbal medicines provide a low-cost option for primary health care owing to a lack of conventional health facilities, a symbol of safety, cultural practices, choices, and efficacy. Also recognizing the significance of herbal medicine, the World Health Organization has developed policies, guidelines, and strategies for plant-derived drugs [144].

## 9. The Role of Plants in Modern Medicine

Medicinal and aromatic plants are excellent resources for creating novel medicines and treating the body and mind [145,146]. The administration of plants, aromatherapy,

crude medications, and a variety of other treatments has become popular not only in contemporary medicine but also in the home and the hospital [138,147]. These natural medications and therapies are also effective for preventative medicine. The prospects of medicinal and aromatic plants provide an ever-present hope for human survival [146]. Due to the inability of alternative drug development approaches to offer many chemical constituents in critical therapeutic applications, such as metabolic and infectious illnesses, natural products remain the emphasis of pharmaceutical investigation [148].

Plant-derived natural product investigation continues to offer a unique template for leading chemical discoveries in the biopharmaceutical industry. Natural products have been and will continue to be significant sources of novel pharmacological therapeutic agents. Drug metabolites derived from synthetic drugs have fewer therapeutic benefits and adverse side effects [149,150]. Therefore, medicinal drugs derived from natural sources may be free of side effects because they exert pharmacological and physiological actions inside live cells.

Moreover, natural products have unique molecular characteristics that distinguish them from manufactured molecules. Compared to manufactured items, raw materials often exhibit relatively little structural variety, showcasing distinct chemical structures that have developed gradually. In addition, these compounds often have reduced partition coefficients, showing their inclination toward aqueous environments. Natural products often exhibit specific size ranges, making molecular mass a notable characteristic [151]. Examples of natural compounds that demonstrate these characteristics include secondary metabolites such as alkaloids, flavonoids, and terpenoids [152]. Another example consists of cellulose nanofibers, which are derived from natural cellulose and possess a low linear expansion coefficient. In order to thrive in these areas, organisms have developed and obtained various tactics, resulting in the creation of a distinct and varied range of organic compounds that may serve as models for the design of novel medicinal substances [153,154]. In addition, natural products interact more with enzymes, proteins, and other biological molecules [155]. Furthermore, natural products contain fewer heavy metals and have higher molecular stiffness than synthetic chemicals and combinatorial libraries [148].

Numerous recently discovered drugs are sourced from medicinal plant sources [156]. The medicinal use of plants varies from crude preparations or extracts to refined extracts and species with a single molecular structure. Indeed, various phytomedicines or plant extracts are now being tested in clinical trials to treat a wide range of illnesses. These plant-based remedies have been utilized in traditional medicine for millennia and are currently being researched for their therapeutic potential in contemporary treatment. Cannabidiol (CBD), produced from the cannabis plant, is one example of a plant-derived medicine that is now being tested in clinical trials [157]. CBD has been proven to have anti-inflammatory, analgesic, and anti-anxiety characteristics, and it is being researched for its potential use in the treatment of a range of illnesses, including epilepsy, anxiety, and chronic pain [158]. Artemisinin, a chemical produced from the sweet wormwood plant, is another example being researched for its potential to cure malaria. Artemisinin has been used for millennia in traditional Chinese medicine and is currently being investigated for its usefulness in treating drug-resistant types of malaria [159]. Recent global licenses of numerous novel medications derived from plants and synthetic and semi-synthetic pharmaceuticals based on plant ingredient compounds indicate that medicinal plants remain critical suppliers of new drugs [160,161].

Morphine, an analgesic used in ancient Mesopotamia, was extracted from the *Papaver somniferum* L. in 1816 by the German chemist Serturner [162]. Taxol, a cancer-fighting taxane diterpenoid produced from *Taxus brevifolia* Nutt., was recently licensed to treat refractory ovarian cancer in the United States [122,163]. The French pharmacists Caventou and Pelletier extracted quinine, an antimalarial drug, from the bark of *Cinchona officinalis* L. in 1820 [164]. In addition, nitisinone is a synthetic derivative of the biological substance leptospermone from *Callistemon citrinu* Skeels that is employed to treat tyrosinemia, and tiotropium, an atropine compound derived from *Atropa belladonna* L.,

is effective in treating chronic obstructive pulmonary disorder [19,165]. Delta-9-tetrahydrocannabinol is synthesized from *Cannabis sativa* L. Some of its synthetic analogs were recently licensed in the United States to treat chemotherapy-induced nausea [166,167]. Cannabinoids are also being researched to treat neurological diseases and glaucoma and for use as antiasthmatics, antihypertensives, and potent analgesics [168–170].

In addition to the active ingredients indicated above, numerous phytochemical ingredients from plants are being studied for their possible benefit (Table 1). For instance, the medicinally active compounds vitamin E, beta-carotene, and ellagic acid are being tested and assessed for their potential role as prototype cancer-preventing and antimutagenic agents [83,171,172]. Furthermore, the organosulfur ingredients of onions and garlic are being researched and assessed for potentially valid cardiovascular mechanisms [168,173–175].

**Table 1.** The main plant-derived natural products and their medicinal applications.

| Plant-Derived Natural Products | Botanical Source | Medicinal Application | Reference |
|---|---|---|---|
| Atropine | *Atropa belladonna* L. | Anticholinergic | [176] |
| Berberine | *Berberis vulgaris* L. | Bacillary dysentery | [177] |
| Caffeine | *Camellia sinensis* (L.) Kuntze | Neuroprotection | [178] |
| Camptothecin | *Camptotheca acuminata* Decne. | Anticancer | [179] |
| Cocaine | *Erythroxylum coca* Lam. | Anesthetic | [180] |
| Colchicine | *Colchicum autumnale* L. | Antigout, antitumor | [181] |
| Convallatoxin | *Convallaria majalis* L. | Cardiotonic | [182] |
| Digitoxin | *Digitalis purpurea* L. | Cardiotonic | [183] |
| Digoxin | *Digitalis lanata* Ehrh. | Cardiotonic | [184] |
| Ephedrine | *Ephedra sinica* Stapf | Sympathomimetic | [185] |
| Glaucine | *Glaucium flavum* Crantz | Antitussive | [186] |
| Glycyrrhizin | *Glycyrrhiza glabra* L. | Treatment of Addison's disease | [187] |
| Morphine | *Papaver somniferum* L. | Analgesic | [188] |
| Ouabain | *Strophanthus gratus* (Wall. & Hook.) Baill. | Cardiotonic | [189] |
| Quinine | *Cinchona officinalis* L. | Antimalarial | [190] |
| Reserpine | *Rauvolfia serpentina* (L.) Benth. ex Kurz | Antihypertensive | [191] |
| Salicin | *Salix alba* L. | Analgesic | [192] |
| Scopolamine | *Datura metel* L. | Sedative | [193] |
| Silymarin | *Silybum marianum* (L.) Gaertn. | Antihepatotoxic | [194] |
| Taxol | *Taxus brevifolia* Nutt. | Anticancer | [71] |
| Theophylline | *Theobroma cacao* L. | Diuretic | [195] |
| Thymol | *Thymus vulgaris* L. | Topical antifungal | [196] |
| Vinblastine | *Catharanthus roseus* (L.) G.Don | Anticancer | [197] |
| Vincristine | *Catharanthus roseus* (L.) G.Don | Anticancer | [197] |
| Yuanhuacine | *Daphne genkwa* Siebold & Zucc. | Abortifacient | [198] |

## 10. Modern Approaches in the Field of Drug Discovery

Modern pharmaceutical research is progressively shifting toward the exploration of compounds produced by plants due to their medicinal capabilities. The use of biotechnology and synthetic biology techniques to create these compounds in regulated conditions represents significant progress in this domain [199]. This method improves the effectiveness and safety of using these substances because it enables the creation of uncontaminated substances in substantial amounts.

The use of phytochemicals in contemporary medicine is well known because several pharmaceuticals and therapies now employed are derived from natural substances. The combination of biotechnology and synthetic biology has provided new opportunities for discovering and developing drugs [200]. This allows for the production of unique chemicals that have improved therapeutic effects. This technique can transform the pharmaceutical business, resulting in the creation of novel and powerful therapies for a diverse array of illnesses and ailments [201]. In addition, this novel strategy tackles ecological problems by minimizing the need to extract whole plants, a practice that often leads to overexploitation and habitat degradation. Bioreactors facilitate the growth of plant cells or microorganisms

in controlled environments, hence increasing the production of targeted substances [202]. Furthermore, metabolic engineering and pathway engineering techniques may be utilized to change the genetic composition of these organisms, thus enhancing their efficiency in producing desired substances [203]. These innovations not only improve sustainability but also guarantee the uniformity and excellence of the obtained plant-derived chemicals.

Simultaneously, high-throughput screening, computer modeling, and bioinformatics are profoundly transforming the discovery process. High-throughput screening is a process that uses automation to test a massive quantity of plant extracts or isolated compounds against specified biological targets [204,205]. This allows the quick discovery of active compounds. Computational modeling techniques, such as molecular modeling and virtual screening, are used to predict the interactions between compounds originating from plants and biological targets. Molecular modeling is a computational method that uses computer simulations to forecast the behavior of molecules and their interactions with other molecules [206]. It is used for the examination of the arrangement and operation of proteins, DNA, and other biological substances, as well as for the development of novel medications and therapeutic agents [207]. Virtual screening is a computer method used to uncover possible drug candidates from extensive collections of chemical compounds. The process entails using diverse molecular modeling and docking simulations to assess the binding affinity between small compounds and target proteins [208]. This methodology has been used to discover novel lead compounds for pharmaceutical development, as well as to enhance the efficacy of already existing medications [143]. This aids in the identification of prospective therapeutic candidates and the understanding of structure–activity connections. Bioinformatics provides extensive data on the genetic, metabolic, and pharmacological characteristics of these molecules, facilitating the identification of novel compounds and offering valuable knowledge on their biosynthesis processes [209,210]. Technology integration enhances the identification and improvement of plant-based molecules, leading to a more efficient and successful age in drug development.

These modern approaches embody a potent combination that not only addresses environmental issues but also expedites the discovery and development of novel pharmaceuticals. Biotechnology and synthetic biology enable the precise manufacture of plant-derived molecules, reducing the impact on the environment and the amount of reliance on natural plant resources [211,212]. Simultaneously, high-throughput screening and computational tools provide a rapid and systematic approach to finding prospective therapeutic compounds, reducing time and resource consumption [213]. Bioinformatics enhances our comprehension of the chemicals, aiding the identification of new pharmaceuticals [214]. In general, the combination of biotechnology and powerful computational tools drives the pharmaceutical sector towards a more environmentally friendly and faster medication development process.

## 11. Future Prospects

Plant-derived natural compounds are positioned to retain their importance as a viable source for developing innovative medications and treatment methods. The extensive range of chemical variations found in plants offers a wide selection of chemicals that have potential uses in medicine [55,67]. As scientists find new bioactive substances and understand the underlying mechanisms of action found in plants, the pathway for drug development remains strong. This is especially crucial when researchers explore alternative and sustainable sources for medications. The progress of investigating and cultivating plant-based substances is closely connected to improvements in genomics and metabolomics. Plant genome sequencing enables scientists to acquire a thorough understanding of the genetic pathways that are accountable for the production of bioactive chemicals [215]. Metabolomics provides a complete experience of the whole range of small molecules present in plant systems. These advanced methods enable scientists to identify prospective chemicals precisely, predict their functions, and modify plant genomes to increase the synthesis of lucrative metabolites. This methodical technique simplifies the process of

finding new information, reducing the need for thorough examination while allowing the development of compounds customized for specific therapeutic characteristics.

Biotechnology and synthetic biology will remain pivotal in advancing the usage of plant-derived compounds. Advancements in bioproduction, genetic engineering, and route optimization are expected to expand the range of substances that may be extracted from plants [216]. Bioreactors, which are enclosed settings, provide a sustainable and scalable method for producing plant-derived molecules while also addressing ecological issues. Genetic alterations and pathway engineering will enable the tailoring of plant metabolites to meet the specific requirements of medication development [217,218]. The future shows potential for incorporating plant-derived chemicals into precision medicine and individualized therapeutics. Using genomic and metabolomic data, researchers will be able to tailor therapies based on an individual's genetic makeup and health characteristics [219]. This method will include choosing or creating certain plant-based substances tailored to the individual health needs of each patient, enhancing the efficacy of treatments while reducing adverse effects. Investigating the combined effects of plant-derived chemicals and their interaction with traditional medications is a rapidly growing field of research. The combination of natural plant chemicals with synthetic pharmaceuticals has the potential to enhance treatment results, reduce side effects, and overcome medication resistance in a wide range of medical problems [12]. The future of drug development from plants may include a deeper investigation of plant species that have yet to be extensively studied [220]. Various regions throughout the globe have distinct and undiscovered biological riches. Scientific research on these animals may reveal previously undiscovered chemicals with remarkable medicinal promise.

## 12. Challenges and Limitations

The excessive extraction of plant species for their therapeutic ingredients might result in the exhaustion of natural resources and could threaten certain plant species. Furthermore, the act of unsustainably collecting resources may lead to habitat degradation, which can cause disturbances in ecosystems and pose a danger to biodiversity. These problems emphasize the need for sustainable and accountable sourcing methods, as well as the preservation of medicinal plant species to save both the ecosystem and indigenous populations reliant on these resources. Ensuring the constant quality and effectiveness of plant-derived chemicals may be a complicated task. Plant genetics, ambient circumstances, and harvesting techniques may influence the chemical makeup of natural products [221]. The standardization of plant extracts, a crucial aspect for ensuring the consistent replication of medicinal actions, might need to be improved. Implementing stringent quality control procedures, such as thorough testing and the setting of precise quality standards, is vital to ensure the safety and effectiveness of goods produced from natural sources. Variations in the concentration of bioactive substances or impurities affect the dependability and safety of herbal remedies. The pharmacokinetics of plant-derived substances exhibit significant variability, which impacts their bioavailability, distribution, metabolism, and elimination within the human body [222]. The efficiency of pharmacological medicines may be influenced by their absorption rates, interactions with other compounds, and chemical stability. Comprehending and enhancing the pharmacokinetic characteristics of plant chemicals is essential for developing medications that exhibit predictable and consistent therapeutic effects [223]. This may include implementing chemical alterations, using formulation methods, or creating drug delivery systems to enhance the efficacy of these substances in clinical environments. The regulation of plant-derived medications may be complex, with variances in rules across nations and regions. Regulatory bodies often mandate stringent safety and effectiveness testing, which results in increased time and expenses throughout the development process [34]. Furthermore, the commercialization of traditional knowledge and plant resources may give rise to concerns about intellectual property, sparking discussions about equitable remuneration for indigenous people and the safeguarding of their cultural legacy. Certain botanical substances are obtained from

distinct geographic areas and may not be easily accessible or economical for all people [224]. The restricted availability of these medicines might impede fair access to their benefits, particularly for those residing in underserved or economically disadvantaged regions. To tackle these obstacles and restrictions, it is necessary to adopt a multidisciplinary strategy that encompasses conservation initiatives, sustainable harvesting procedures, stringent quality control standards, regulatory harmonization, and ethical concerns. Utilizing plant-derived chemicals in drug discovery and development presents significant opportunities. Still, it is crucial to approach these challenges responsibly to optimize benefits and mitigate harmful effects on the environment, society, and healthcare.

## 13. Conclusions

Natural compounds deriving from plants have long been and will continue to be immensely significant as sources of therapeutic agents and models for the design, semisynthesis, and synthesis of various drugs for managing human and animal ailments. With increased interest in developing herbal medications with few adverse effects, there are more chances to investigate the therapeutic and other biological aspects of previously unexplored natural items. In plant-derived drug development and discovery investigations, plant phytochemicals are optimized to create possible analogs with the requisite effectiveness and safety. Due to medicinal chemists' increasing interest in natural substance drug development, various innovative techniques and technological advances have been created for natural ingredients' selection, identification, isolation, characterization, and biological screening. These innovative techniques might decrease the technological disadvantages associated with natural product development and overcome the difficulties experienced in discovering and producing novel natural remedies due to their complicated behavior. It is envisaged that plants will continue to yield unknown biomolecules, allowing the discovery of unique, innovative remedies for microbial infections and diseases. Additionally, the increasing demand for medicinal plants in herbal medicine and drug discovery and development threatens their survival. Consequently, it is critical to ensure that threatened, vulnerable, and overexploited genetic resources are maintained to the maximum degree feasible for future generations who will have the capabilities to manage and exploit these species more adequately and sensibly.

**Author Contributions:** Conceptualization, N.C.; methodology, N.C.; software, N.C.; validation, L.Z.; formal analysis, N.C.; investigation, N.C.; resources, N.C.; data curation, N.C.; writing—original draft preparation, N.C.; writing—review and editing, L.Z.; visualization, L.Z.; supervision, L.Z.; project administration, N.C.; funding acquisition, N.C. Both authors have read and agreed to the published version of the manuscript.

**Funding:** This research received no external funding.

**Data Availability Statement:** No new data were created or analyzed in this study. Data sharing is not applicable to this article.

**Conflicts of Interest:** The authors declare no conflicts of interest.

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
