# Peer review of "Plant-Derived Natural Products: A Source for Drug Discovery and Development"

_ddc, doi:10.3390/ddc3010011_

Round 1
Reviewer 1 Report
Comments and Suggestions for Authors
Comments:
Introduction should be enlarged discussing previous review on the topic.
All manuscript needs more in-depth discussion, including more examples and approaches. In-depth discussion on the secondary metabolites and the use of plant-derived products in the Traditional Pharmacopeias are needed.
Author Response
Dear Prof.
Please see attached document

Reviewer 2 Report
Comments and Suggestions for Authors It a an review article that try summarizes an overview the plant-derived natural products as drug discovery and candidates in drug development. The idea is fine, however in many part the review is too general. This is my main critical concern. Therefore, I recommend to provide more details to this review. Minor comments The Figure 1 is informative and good designed. Table 1 Berberis vulgaris – should be Berberis vulgaris L. As in chapter 3 you gave a historical overview , the 3. 3 and 3.4 should be over changed; i.e give Quinine: A Malaria Breakthrough from Cinchona (17th Century) before 3.3. Aspirin: A Gift from the Willow Tree (19th Century); his will allow you to maintain the chronological order.L 70 – you declared the review of 97 papers, meanwhile in the Reference list I found 165 papers????
L 165 Give some more detailed information
Scientists can use the innate protective mechanisms of plants to create medications that specifically target certain illnesses or situations [44].
L 293 = you mentioned the report; what report; the reference is needed
L 305-311 be more consequent; for polyphenols the examples are given; give the examples also for flavonoids; fruits vegetables – it very broad; be more precise
L 313-317 – no references at all. ; and very general statement is given. Present more details.
L 358 -359 Moreover, natural products exhibit a broader spectrum of molecular attributes, such as reduced structural diversity, partition coefficient, and molecular mass [103, 104].
Again, it is very broad statement; give some examples
L 455-468 – no references; it is required in review article.
Author Response
Dear Prof.
Please see attached document

Reviewer 3 Report
Comments and Suggestions for Authors
In review article entitled “Plant-Derived Natural Products: A Source for Drug Discovery and Development” current advancements and future approaches for discovering natural items such as health and wellness-promoting remedies were discussed. Due to medicinal chemists' increasing interest in natural substance drug development, various innovative techniques and technological advances have been created for natural ingredients' selection, identification, isolation, characterization, and biological screening.
The paper is interesting, the methodology is adequate and explicitly stated, the subject is very topical and for this reason, I recommend the publication of this study.
Author Response
Dear Prof.
Please see attached document

Reviewer 4 Report
Comments and Suggestions for Authors
The review is well organized but check the labeling of different chapter (page 7 from 5.6 to 3 ?) In the 5.2 extraction part You described very quickly and with old method! Now there are ecoextraction methods and new biobased solvants.
Author Response
Dear Prof.
Please see attached document

Round 2
Reviewer 1 Report
Comments and Suggestions for Authors
The authors have changed their manuscript according to the reviewer`s recommendation.
Reviewer 2 Report
Comments and Suggestions for Authors
My comments have been properly addressed
Reviewer 4 Report
Comments and Suggestions for Authors
The manuscript is more understandable and better described the differents aspects of Drug design using natural compounds. Please recheck the number in chapter 3 ! chapter 6 you could cite the Nagoya protocol for protection of biodiversity.